# Traditional Uses, Phytochemical Composition, Pharmacological Properties, and the Biodiscovery Potential of the Genus *Cirsium*

Gaurav Aggarwal [1] , Gurpreet Kaur [2], Garima Bhardwaj [3], Vishal Mutreja [4], Harvinder Singh Sohal [4], Gulzar Ahmad Nayik [5] , Anikesh Bhardwaj [4] and Ajay Sharma [6,*]

1 Council of Scientific & Industrial Research—Institute of Himalayan Bioresource Technology, Palampur 176061, India
2 Department of Zoology, Mata Gujri College, Fatehgarh Sahib 140407, India
3 Department of Chemistry, Sant Longowal Institute of Engineering and Technology, Sangrur 148106, India
4 Department of Chemistry, Chandigarh University, Mohali 140413, India
5 Department of Food Science & Technology, Govt. Degree College Shopian, Srinagar 192303, India
6 University Centre for Research & Development, Department of Chemistry, Chandigarh University, Mohali 140413, India
* Correspondence: sharmaajay9981@gmail.com

**Abstract:** Medicinal plants are rich in phytochemicals, which have been used as a source of raw material in medicine since ancient times. Presently they are mostly used to treat Henoch–Schonlein purpura, hemoptysis, and bleeding. The manuscript covers the classification, traditional applications, phytochemistry, pharmacology, herbal formulations, and patents of *Cirsium*. The main goal of this review is to impart recent information to facilitate future comprehensive research and usthe e of *Cirsium* for the development of therapeutics. We investigated numerous databases PubMed, Google Scholar, Springer, Elsevier, Taylor and Francis imprints, and books on ethnopharmacology. The plants of the genus *Cirsium* of the family Asteraceae contain 350 species across the world. Phytochemical investigations showed that it contains flavonoids, phenols, polyacetylenes, and triterpenoids. The biological potential of this plant is contributed by these secondary metabolites. *Cirsium* plants are an excellent and harmless agent for the cure of liver diseases; therefore, they might be a good clinical option for the development of therapeutics for hepatic infections. The phytochemical studies of different *Cirsium* species and their renowned pharmacological activities could be exploited for pharmaceutic product development. Furthermore, studies are required on less known *Cirsium* species, particularly on the elucidation of the mode of action of their activities.

**Keywords:** *Cirsium*; medicinal plants; secondary metabolites; phytochemistry; pharmacology; toxicology

## 1. Introduction

In recent years, there has been a surge in interest in natural products for the prevention and cure of many diseases such as cancer, arthritis, cardiovascular disorders, and diabetes. Plants and their natural products are reservoirs of phytoconstituents that have antimicrobial, antioxidant, anticancer, and antidiabetic properties and are used in traditional medicine [1]. Plant extracts provide limitless opportunities for novel medicinal discoveries due to their unrivaled chemical variety and biocompatible nature. Many studies are being conducted to identify an alternative therapy using medicinal plants. In reality, roughly 25% of the medications on the market are derived directly or indirectly from plants [2]. Many of these plants have recently been suggested for their capacity to act as a preservative and food additive, serving a dual function of culinary taste and bioactive constituent [3].

*Cirsium* is a genus of perennial and rarely annual prickly Asteraceae plants. It gets its name from the Greek word "khirsos," which means "swollen vein." According to the Plants of the World online collections at the Royal Botanic Gardens of Kew, there are around 450–480 recognized species in this genus [4,5]. These plants are found in the northern

hemisphere, including Eurasia, Asia, North America, and North Africa. There are around 120 and 50 *Cirsium* species in Japan and China, respectively [6,7]. The *Cirsium* has been used traditionally as a folk medicine for the treatment of various ailments in India and in many neighboring countries such as China for centuries. Many species of *Cirsium* are exploited as an herbal remedy for the cure of cardiovascular diseases and used as anti-inflammatory and diuretic agents traditionally. *Cirsium* is still employed clinically as cold blood hemostatic medicine. Pharmacological research studies have revealed that *Cirsium* is valued for its wide range of therapeutic benefits, such as hepatoprotection, antioxidant, anticancer, antimicrobial, and anti-inflammation, which may untie the multiple medicinal applications of *Cirsium* [8]. Moreover, some *Cirsium* species have been used for the cure of diabetes, nervousness, and overweightness. *Cirsium* is a multifunctional herb that is utilized for the cure of hemoptysis, hematuria, distressing bleeding, and Henoch–Schonlein purpura. Phytochemical investigations showed that the *Cirsium* species contained flavonoids, polyacetylenes, acetylenes, phenolic acids, phenylpropanoids, sterols, and terpenoids [9]. Among them, flavonoids, phenylpropanoids, and terpenoids are considered to be the main phytocompounds and are accountable for a large number of biological properties found in the different species of this genus. Variation in pharmacological activity is attributed to the different chemical compositions in different *Cirsium* species from different regions [10,11]. This review summarizes and evaluates the existing traditional uses, botanical description, phytochemical composition, pharmacokinetics, and biological properties of the *Cirsium* genus. Furthermore, the progress of research on herbal formulations, patents, and the safe profile of *Cirsium* is also discussed. This review article represents updated information on *Cirsium* to benefit future scientific investigations for the production of novel drug formulations and clinical trials.

## 2. Methodology

This review article was designed by collecting relevant information about the genus *Cirsium* from various databases, i.e., PubMed, Google Scholar, Chem Spider, Springer, World Scientific, Science Direct Elsevier, Taylor and Francis imprints, peer-reviewed journals, and some of the unpublished data. In addition, several 'grey literature' sources, such as Wikipedia, online sites, ethnobotanical books, and chapters, are included in the data sources.

## 3. *Cirsium*

There are nearly 450 *Cirsium* species found across the world [8]. Around 200 of them are found in various regions of Asia, Central America, Europe, North Africa, and North America [12]. Out of these species, about 16 known species are found in Indian evergreen forests and some regions bordering Nepal and China also, viz. *C. argyracanthum, C. arvense, C. lineare, C. eriophoroides, C. falconeri, C. flavisquamatum, C. interpositum, C. verutum, C. nishiokae, C. phulchokiense, C. shansiense, C. souliei, C. tibeticum, C. verutum, C. wallichii* and *C. glabrifolium* [13]. *Cirsium* originated in Eurasia and Northern Africa, and roughly 60 species have been identified from North America. It can be found in every continent except Antarctica, albeit its range is largely limited to the northern and southern temperate regions [14–16] (Table 1). *Cirsium* genus comes under the Asteraceae family (Figure 1), and this family consists of about 1600 genera and 23,000 species [15]. Most of the species of this family are perennial thistles and have spines on leaves, flowers, stems, and roots or some time on the whole plant. Species of genus *Cirsium* are entitled 'common thistle', and 'field thistle' (Table 1). More precisely, they are termed 'Plume thistles' since some genera, i.e., *Carduus, Silybum,* and *Onopordum,* are frequently labeled by the term 'thistle' [16]. *Cirsium* is considered a weed that grows anywhere, including near farmed areas. Despite their status as weeds, they have the potential for allelopathic impact and biological pest control [17]. Some species are even planted in gardens owing to their aesthetic appeal [18,19].

**Table 1.** Distribution and pharmacological potential of some species of *Cirsium*.

| S. No | Species of *Cirsium* | Geographical Distribution | Traditional Uses | Common Name | Major Phytoconstituents | Pharmacological Applications | References |
|---|---|---|---|---|---|---|---|
| 1 | *Cirsium arvense* | Europe, Asia, Northern Africa, India | Pharyngitis, astringent, tonic, tumor, diuretic, toothache, diaphoretic | Canada thistle, Creeping thistle, Field thistle, Californian thistle | Acacetin, Ciryneol C, Hispidulin, Pectolinarigenin, Luteolin, Tracin, Scopoletin, Apigenin, Citronellol | Antimicrobial, antifungal, anticancer, antidiabetic, neuroprotective, anti-inflammatory | [5,8] |
| 2 | *Cirsium oleraceum* | Europe to West Siberia and Kazakhstan | Anxiolytic, diuretic, astringent, antiphlogistic, Antitumor | Cabbage thistle, Siberian thistle | Thymol, Carvacrol, Luteolin, Apigenin, Methylkaempferol | Antioxidant, antimicrobial, anti-glioma effect | [20] |
| 3 | *Cirsium englerianum* | Ethiopia | Dermal infections, cough, snake bite, hematuria, diarrhea anthrax, anti-scabies | - | Alkaloids, Quinones, Terpenoids, Phenolics, and Flavonoids | Antioxidant, antimicrobial | [5] |
| 4 | *Cirsium eriophorum* | China South-Central, East Himalaya, Myanmar, Tibet | Detoxification and cure of hepatic infections | Woolly thistle | Vanillic acid, Balanophonin, Apigenin, Kaempferol, Taraxasterol, Sitosterol, Linoleic acid | Antioxidant, acetyl-cholinesterase inhibitory activity | [21] |
| 5 | *Cirsium wallichii* | Afghanistan, East Himalaya, Nepal, Pakistan, West Himalaya | Pyrexia, bleeding relief, burning sensation, and stomach inflammation | Wallich's Thistle, Plume thistles | Acetyljacoline, Fumaric acid | Antimicrobial, antifungal, antioxidant | [5] |
| 6 | *Cirsium verutum* | Assam, East Himalaya, Myanmar, Nepal, Pakistan, Tibet, Vietnam, West Himalaya | Typhoid, bleeding, chest pain, measles, purgative, pharyngitis, dyspepsia, dysentery, tuberculosis | Common thistle, Creeping thistle, Plume thistle | Lupeol, Taraxasterol acetate, Pectolinarigenin, Cirsitakaoside, Cirsitakaogenin, Pectolinarin | Antimicrobial, antifungal | [22] |
| 7 | *Cirsium setidens* | Korea | Pyrexia, detoxify, and improve blood circulation | Ungungqwui' in Korea, Thistles in English | Linarin, Phytol, Syringin, Pectolinarigenin, Cyclocitral, Pentylfuran, Trans-β-Ionone Rutin, Setidenosides, Isorhamnetin | Antimicrobial, antifungal, anticancer, neuroprotective, anti-inflammatory Antidiabetic, osteogenic agent | [23] |
| 8 | *Cirsium tenoreanum* | Italy | Treatment of varicose | Cardo di Tenore | Kaempferol, Apigenin, Quercetin-3-*O*-galactoside | Antimicrobial, antiproliferative | [24] |
| 9 | *Cirsium vulgare* | Europe to Siberia and Arabian Peninsula, West Himalaya | Anxiolytic | Spear or bull thistle | Quercetin, Apigenin, Kaempferol, and Luteolin | Antioxidant, antimicrobial | [9] |
| 10 | *Cirsium japonicum* | China, Korea, Japan | Hemorrhages, cancer, hypertension, and hepatitis | Japanese thistle | Linarin, Luteolin, Coumaric acid Pectolinarin, Ciryneol, Syringin, Cirsimaritin Pectolinarigenin, Lupenyl acetate | Anticancer, anti-Alzheimer. anti-inflammatory, antimicrobial | [25] |

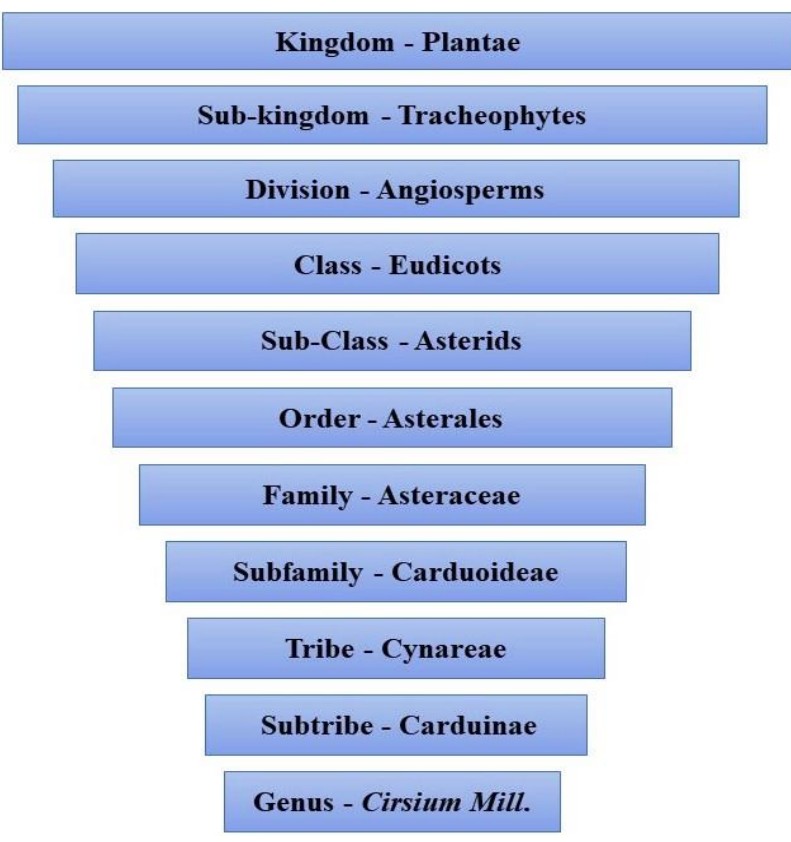

**Figure 1.** The botanical classification of the genus *Cirsium*.

## 4. Botanical Description of Genus *Cirsium*

Thistles are recognized by their demonstrative flower heads (purple, rose, pink, yellow, or white) with many tiny flowers. Flowers are radially symmetrical disc-like structures blooming at the end of the branches. Their flowers bloom from April to August [9]. Plant constitutes straight stems and prickly leaves having a distinctive enlarged base of flower which are usually spiny; leaves are alternate, some species can be less or more hairy; leaf extension leading to the stems called wings, can be conspicuous (*Cirsium vulgare*), lacking, or inconspicuous (Figure 2). They can reproduce by pollination and also through rhizomes (*Cirsium arvense*) and are annual, biennial, and some are perennial in growth form [10,26].

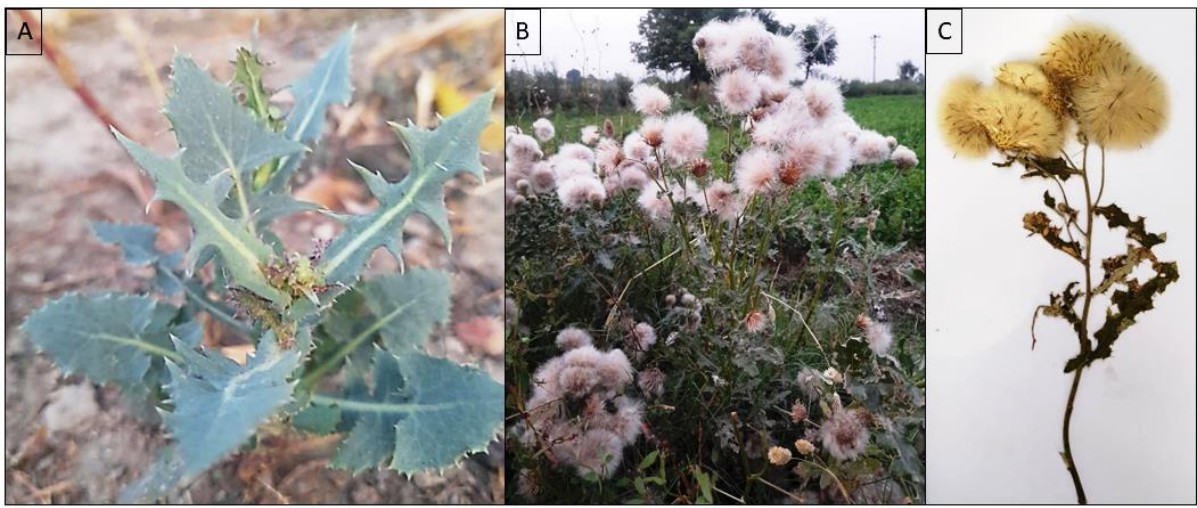

**Figure 2.** (**A**) Leaves of *Cirsium* (**B**) *Cirsium* Whole Plant (**C**) Aboveground parts of *Cirsium*.

## 5. Traditional Uses

Currently, traditional plants have emerged as a viable avenue for the discovery of novel therapeutic molecules to treat a variety of severe ailments, such as diabetes and cancer [27]. More than 10 species of *Cirsium* are well known in Chinese medicines to cure jaundice, hemorrhaging, gastrointestinal issues, and vascular diseases [28]. Traditionally, *Cirsium* is also exploited as an herbal remedy for the healing of leukemia and peptic ulcers [29]. In Central America and Mexico, *C. mexicanum* is utilized in traditional medicine for the cure of respiratory problems, hepatic infections, diarrhea, dysentery, and stomach pain [23]. Moreover, *Cirsium rivulare* is utilized traditionally in Poland for anti-anxiety effects [30]. The fruits and roots are also used to cure constipation, dyspepsia, skin problems, chest discomfort, and as a tonic by the people of the Himachal region [31]. The indigenous culture of Meghalaya uses leaf extract to treat gastrointestinal ailments, primarily diarrhea, and dysentery. The Bhotiya tribal community of central Himalaya, India, uses the *Cirsium* the cure rheumatism [32]. *Cirsium arvense* roots are diuretic, astringent, antiphlogistic, and hepatoprotective. Root decoction of *Cirsium* has been traditionally employed for treating worm infection in children. Moreover, root paste of *Cirsium arvense* in combination with *Amaranthus spinosus* is used for the treatment of indigestion. Rheumatic joint pains have long been treated using a hot brew made from *Cirsium arvense*. Decoction of the whole plant has been used as a curative medicine internally and externally for bleeding piles [33]. It has been surveyed in Central Italy that *Cirsium arvense* leaf soup is used to alleviate digestive problems and stomach discomfort. Moreover, *Cirsium arvense* is also utilized to halt the flow of blood from wounds in an emergency [34]. Extract and infusions of *C. arvense* leaves are a good repository of fibers, vitamins, and important minerals, and these were used by North American Indians for toothaches, tuberculosis, throat sores, and cancer sores due to their potent anti-inflammatory properties [35]. Other species, such as *Cirsium rivulare*, have also been used traditionally for anxiety-related issues [36] and have been documented to exhibit antimitotic action [37]. Moreover, in Polish medicine, *C. arvense* and *C. oleraceum* have been reported to be used as a diuretic, antiphlogistic, and astringent [38]. *C. rivulare*, *C. oleraceum,* and *C. vulgare* have long been recognized for their anxiolytic properties in Poland's Northeastern regions [39]. Aerial portions of *C. chanroenicum* have been reported to be utilized in Chinese medicine to cure pyrexia, detoxify, and improve blood circulation [40]. In addition to it, *C. japonicum* has been utilized in Chinese medicine as a hypertensive, anti-hemorrhagic, and anti-hepatitis agent [25], as well as in folk medicine for the cure of malignancies such as uterine and liver cancer [41]. The water extract of *C. arisanense* displayed a hepatoprotective effect and has been applied in Taiwanese traditional medication for hundreds of years [42]. In another work, *Cirsium* species seeds and root decoction have been employed in Turkish traditional medicine to cure hemorrhoids. Flowers of *Cirsium* species ameliorated peptic ulcers, and stems are used for the cure of cough and bronchitis in Anatolia [43]. Root paste of *C. falconeri* and *C. verutum* is useful for the cure of arthritis [22]. Moreover, the dried powder of flowers and leaves of *C. falconeri* has been found to mediate protective effects against Cerebral edema in the Himalayan province of India [44]. The stems and young leaves of *C. setidens* are edible in nature and high in calcium, proteins, and vitamin A. This species of *Cirsium* has been used to treat emesis, hypertension, and hematuria in Korean medicines [45]. Moreover, *Cirsium vulgare* is persistently grown in gardens nowadays to attract birds and butterflies. Hummingbirds are attracted to the flowers of *C. vulgare*, and many immigrants keep hummingbirds in their botanical gardens [16].

## 6. Culinary Uses of *Cirsium*

The leaves, roots, and stem of *Cirsium* are edible and can be consumed raw or cooked. This plant can be utilized in salads or in combination with other vegetables traditionally. The stem of *Cirsium* can be peeled and roasted like Asparagus. In several European countries, the leaves of a few *Cirsium* species are used to make tea, which displays tremendous medicinal properties [36,46–48]. The leaves of *Cirsium* have a fairly mild flavor and can be

consumed either raw or cooked. The intake of boiled leaves serves as an effective diuretic and liver tonic. Moreover, a mixture of the soaked leaves and roots of the *Cirsium* is used as a remedy for the healing of neural problems. The flowers and roots of *Cirsium* are employed for the preparation of an infusion for drinking or applying vaginal douches. The seed oil of *Cirsium* is utilized for cooking and also as a lamp oil [49].

## 7. Phytochemical Composition

### 7.1. Flavonoids

Flavonoids are a kind of polyphenol present in medicinal plants that have been shown to have antioxidant and anticancer effects. The various species of the *Cirsium* contain all the categories of flavonoids, i.e., flavones, flavanones, and flavanols [17]. Most of the flavonoids are identified in *C. rivulare*, *C. japonicum*, and *C. arvense*. The chemical investigation recognized major flavones (Figure 3), which are common in many *Cirsium* species were Linarin, Luteolin, Luteolin 7-*O*-β-D-glucoside, Pectolinarin, Apigenin, Apigenin 7-*O*-glucoside, and Hispidulin [16,23,35,46,47,50–64].

**Figure 3.** Commonly found flavonoids in different species of *Cirsium*.

Besides these compounds, the infrequently distributed flavones were also found in *Cirsium* (Figure 4), namely Pectolinarigenin, Acacetin, 5,7-Dihydroxy-6,4'-dimethoxyflavone, Cirsimaritin, Cirsimarin, Rutoside and Tricin [51,53,54,57,59,60,65]. In addition to it, flavones glucoside was also detected in species of *Cirsium* (Figure 5), specifically Pectolinarigenin-7-*O*-glucopyranoside, Hispidulin-7-neohesperidoside, Hispidulin-7-glucoside, Luteolin 7-*O*-β-D-glucuronide, Isokaempferide-7-glucuronide, Isokaempferide-7-*O*-β-D-(6″-methylglucuronide), Apigenin 7-(6″-methylglucuronide), and 6-Hydroxyluteolin 7-*O*-glucoside [50,51,56,64]. In another work, Eriodictyol 7-*O*-glucoside [56], Kaempferol, Kaempferol 3-galactoside, Kaempferol 3-glucoside, and Isorhamnetin were detected in *Cirsium* species [50,61,63,66]. Moreover, some polyphenolic phytoconstituents 4-Vinyl guaiacol, 4-Ethyl guaiacol, Scopoletin, and 6,7-Dimethoxycoumarin were also reported in different species (Tables 1 and 2) of *Cirsium* (Figure 6) [17,35,51]. These compounds are responsible for the natural aroma in plants and possess fungicidal properties. They can halt conidium germination and germ tube elongation in several plant pathogenic fungi [32,67].

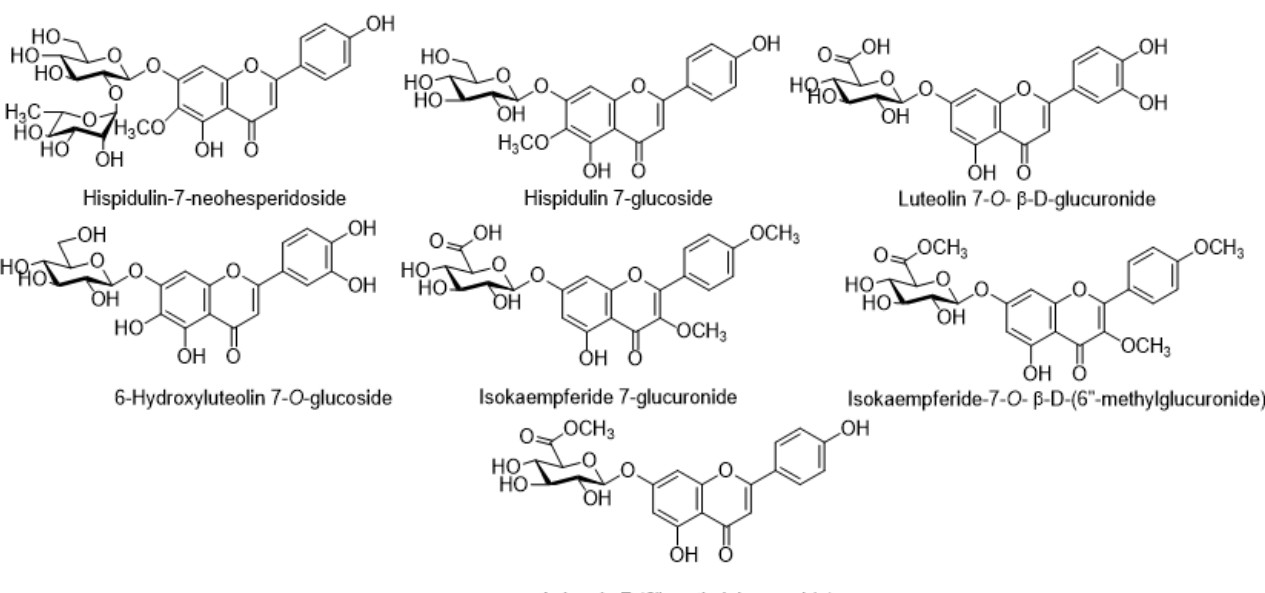

**Figure 4.** Infrequently distributed flavonoids in *Cirsium* species.

**Figure 5.** Major flavone glucoside detected in different species of *Cirsium*.

**Figure 6.** Other polyphenolic compounds isolated from *Cirsium* species.

**Table 2.** Major phytoconstituents present in various species of *Cirsium*.

| Category of Phytoconstituent | Name of Phytoconstituents | Species | References |
|---|---|---|---|
| Flavanoids | Linarin | *C. arvense* | [46] |
| | | *C. japonicum* | [62,64] |
| | | *C. setosum* | [64] |
| | | *C. rivulare* | [65] |
| | | *C. canum* | [63] |
| | Scopoletin | *C. arvense* | [51] |
| | Pectolinarigenin | *C. chanroenicum* | [59] |
| | | *C. setidens* | [60] |
| | Pectolinarigenin-7-*O*-glucopyranoside | *C. arvense* | [51] |
| | Acacetin | *C. arvense* | [51] |
| | 6,7-Dimethoxycoumarin | *C. arvense* | [51] |
| | Tracin | *C. arvense* | [51] |
| | Hispidulin | *C. arvense* | [35] |
| | | *C. japonicum* | [54,55] |
| | | *C. rivulare* | [30] |
| | Hispidulin-7-neohesperidoside | *C. japonicum* | [64] |
| | Luteolin | *C. arvense* | [35] |
| | | *C. japonicum* | [55,64] |
| | | *C. canum* | [63] |
| | | *C. palustre* | [56] |
| | | *C. rivulare* | [62] |
| | Luteolin 7-*O*-β-D-glucuronide | *C. scabrum* | [26] |
| | Luteolin 7-*O*-β-D-glucoside | *C. scabrum* | [26] |
| | | *C. canum* | [63] |
| | | *C. palustre* | [56] |
| | Eriodictyol 7-*O*-glucoside | *C. palustre* | [56] |

**Table 2.** *Cont.*

| Category of Phytoconstituent | Name of Phytoconstituents | Species | References |
|---|---|---|---|
| | | *C. japonicum* | [53,57,64] |
| | | *C. rivulare* | [50] |
| | Pectolinarin | *C. subcoriaceum* | [58] |
| | | *C. chanroenicum* | [59] |
| | | *C. setidens* | [60] |
| | | *C. japonicum* | [25] |
| | Isokaempferide 7-*O*-β-D-(6″-methylglucuronide | *C. rivulare* | [50] |
| | Isokaempferide 7-glucuronide | *C. rivulare* | [50] |
| | | *C. canum* | [63] |
| | Apigenin | *C. setosum* | [61] |
| | | *C. rivulare* | [30] |
| | | *C. japonicum* | [55] |
| | Apigenin 7-(6″-methylglucuronide) | *C. rivulare* | [30,50] |
| | Apigenin 7-glucoside | *C. canum* | [63] |
| | | *C. rivulare* | [30] |
| | Kaempferol | *C. canum* | [63] |
| | Kaempferol 3-galactoside | *C. rivulare* | [50] |
| | Kaempferol 3-glucoside | *C. canum* | [63] |
| | Kaempferol 3- β-D-glucopyranoside (Astragalin) | *C. setosum* | [61] |
| | 4-Vinyl guaiacol | *C. creticum* | [17] |
| | 4-Ethyl guaiacol | *C. creticum* | [17] |
| | 5,7-Dihydroxy-6,4′-dimethoxyflavone | *C. japonicum* | [53,57] |
| | Cirsimaritin | *C. japonicum* | [49] |
| | Cirsimarin | *C. japonicum* | [49] |
| | Rutoside | *C. canum* | [68] |
| | 6-Hydroxyluteolin 7-*O*-glucoside | *C. palustre* | [56] |
| | Tricin | *C. rivulare* | [69] |
| | Isorhamnetin | *C. helenioides* | [66] |
| Steroids | Stigmasterol | *C. arvense* | [46] |
| Steroidal glucoside | Daucosterol | *C. arvense* | [46] |
| Alkaloids | Benzymidazole | *C. arvense* | [46] |
| | α-Tocopherol | *C. setidens* | [23] |
| | | *C. arvense* | [35] |
| Terpenes | α-Tocospiro A, B and C | *C. setosum* | [70] |
| | 4(15),10(14)-Guaiadien-12,6-olide | *C. setidens* | [23] |
| | Trans-Phytol | *C. setidens* | [23,71] |
| | Dihydroactinidiolide | *C. creticum* | [17] |

**Table 2.** *Cont.*

| Category of Phytoconstituent | Name of Phytoconstituents | Species | References |
|---|---|---|---|
| Triterpenes | Lupeol | *C. scabrum* | [47] |
| | Lupeol acetate | *C. palustre* | [72] |
| | Taraxasterol acetate | *C. scabrum* | [47] |
| | 25-Hydroperoxycycloart-23-en-3β-ol | *C. scabrum* | [47] |
| | | *C. setidens* | [23] |
| | β-Amyrin | *C. palustre* | [72] |
| | Faradiol | *C. palustre* | [72] |
| Sesquiterpenes | Caryophyllene oxide | *C.setidens* | [71] |
| | β-Caryophyllene alcohol | *C.setidens* | [71] |
| Cyclic ether | Ciryneol | *C. arvense* | [51] |
| | 1,2,15,16-Diepoxyhexadecane | *C. setidens* | [71] |
| Fatty acids | 9, 12, 15-Octadecatrienoic acid | *C. setidens* | [23] |
| | 9, 12-Octadecadienoic acid | *C. setidens* | [23] |
| | Hexadecanoic acid | *C. setidens* | [23] |
| | | *C. creticum* | [17] |
| | | *C. palustre* | [72] |
| | Palmitic acid | *C. japonicum* | [41] |
| Sterols | Acylglycosyl β-sitosterol | *C. setidens* | [23] |
| | β-Sitosterol glucoside | *C. setidens* | [23] |
| | Taraxasterol | *C. setosum* | [61] |
| Glycerol | Monogalactosyldiacyl glycerol | *C. setidens* | [23] |
| | | *C. helenioides* | [66] |
| | | *C. palustre* | [73] |
| | | *C. rivulare* | [73] |
| | Dihydroaplotaxene | *C. helenioides* | [66] |
| | Tetrahydroaplotaxene | *C. helenioides* | [66] |
| | Pentacosane | *C. setidens* | [71] |
| Aldehydes | Sinapaldehyde | *C. helenioides* | [66] |
| Ketones | 6,10,14-Trimethyl-2-pentadecanone | *C. setidens* | [71] |
| Phenolic acids | Chlorogenic acid | *C. canum* | [63] |
| | | *C. palustre* | [56] |
| | Caffeic acid | *C. canum* | [63] |
| | p-Coumaric acid | *C. canum* | [63] |
| | Protocatechuic acid | *C. canum* | [63] |
| | p-Hydroxybenzoic acid | *C. canum* | [63] |
| | Vanillic acid | *C. canum* | [63] |
| | Syringic acid | *C. canum* | [63] |
| | Trans-Cinnamic acid | *C. canum* | [63] |

### 7.2. Terpenoids and Sterols

The terpenes or terpenoids constitute the largest class of secondary products. They are synthesized by the Mevalonic acid pathway in the chloroplast of the plants. There are various types of terpenes utilized for the treatment of several infectious diseases. According to many studies, terpenes possess very good immunomodulatory properties [74]. The main triterpenes (Figure 7) detected in *Cirsium* species were Lupeol, Lupeol acetate, Taraxasterol acetate, 25-Hydroperoxycycloart-23-en-3β-ol, 24-Hydroperoxycycloart-25-en-3β-ol, α-Amyrin acetate, β-Amyrin acetate, β-Amyrin, and Faradiol [23,72]. In addition to it, other terpenes found in *Cirsium* species were α-Tocopherol, α-Tocospiro, 4 (15),10 (14)-Guaiadien-12, 6-olide (mokko lactone), t-Phytol, and Dihydroactinidiolide [17,23,35,70,71]. Among the sesquiterpenes, Caryophyllene oxide and β-Caryophyllene alcohol [71] were also observed in the *Cirsium*. Moreover, major sterols identified in *Cirsium* species were Acylglycosyl β-Sitosterol, β-Sitosterol glucoside, Taraxasterol, Stigmasterol, and Daucosterol (Figure 8) [23,46,61,72].

**Figure 7.** Terpenoids found in different species of *Cirsium*.

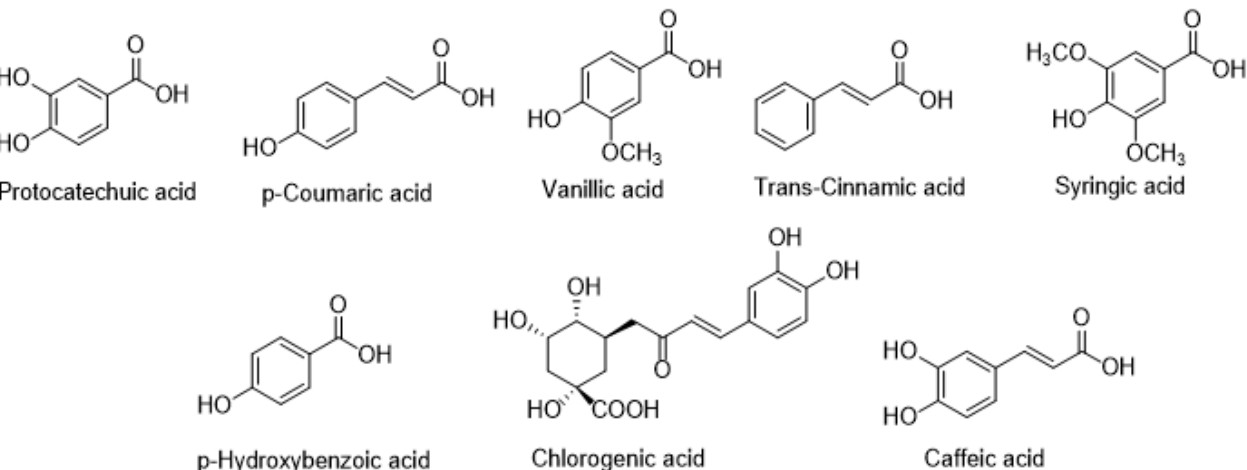

**Figure 8.** Major sterols identified in different species of *Cirsium*.

### 7.3. Phenolic Acids

The phenolic acid (Figure 9) in the different species of *Cirsium* were found to be Protocatechuic acid, Caffeic acid, Vanillic acid, Chlorogenic acid, p-Coumaric acid, p-Hydroxybenzoic acid, trans-Cinnamic acid, Syringic acid, and Caffeic acid [52,56,63,75].

**Figure 9.** Major phenolic acids found in different species of *Cirsium*.

### 7.4. Polyacetylenes, Acetylenes, and Hydrocarbons

The Acetylenes and Polyacetylenes have been proven in studies to be distinctive of the Asteraceae family [75–83]. However, Polyacetylenes and Acetylenes were found in the *Cirsium*, namely Aplotaxene, Dihydroaplotaxene, Pentacosane, Tetrahydroaplotaxene, and 1-Pentadecene [41,66,71,73]. Ciryneol and 1,2,15,16-Diepoxyhexadecane are two hydrocarbons detected in the *Cirsium* (Figure 10). Ciryneol is a cyclic ether that was found in *C. arvense* [51,71]. All these Acetylenes and Polyacetylenes were isolated from the non-aerial part of the *Cirsium* species [16].

**Figure 10.** Polyacetylenes, acetylenes, and hydrocarbons found in *Cirsium* species.

*7.5. Fatty acids, Aldehydes, and Ketones*

The fatty acids identified in the *Cirsium* were 9,12,15-Octadecatrienoic acid, 9,12-Octadecadienoic acid, Hexadecanoic acid, and Palmitic acid [17,23,35,41,71,72]. Other carbonyl compounds (Figure 11) recognized in the *Cirsium* were Sinapaldehyde and 6,10,14-Trimethyl-2-pentadecanone [66,71].

**Figure 11.** Fatty acids, aldehydes, and ketones found in *Cirsium* species.

## 8. Essential Oil Composition

The essential oil was extracted from *C. acaule, C. arvense, C. creticum, C. decussatum, C. dissectum, C. eriophorum, C. heterophyllum, C. japonicum, C. ligulare, C. oleraceum, C. palustre, C. pannonicum, C. rivulare,* and *C. setidens* [65,68,71,73,84–86]. It was observed that essential oil from *Cirsium* was found to be a rich source of Aplotaxene and Hexadecanoic acid. The maximum concentration of Aplotaxene was observed in the *C. japonicum, C. palustre, and C. rivulare.* However, Hexadecanoic acid was found to be more in *C. japonicum, C. creticum,* and *C. setidens* [68,71,73,86]. Other components such as (Z)-8,9-Epoxyheptadeca-1,11,14-triene in *C. palustre* and *C. rivulare* [73]; Pentadecanoic acid, Heptacosane, Heptadecanoic acid, Tetradecanoic acid, Palmitic acid, Caryophyllene oxide, and Myristic acid in rhizomes of *C. japonicum* [65]; Phytol in *C. setidens* and *C. arvense* [71,84]; 4-Ethyl guaiacol, (E)-β-Damascenone and Dihydroactinidiolide in *C. creticum* [17]; α-Bisabolol, δ-Cadinene, Hexacosane, β-Selinene, α-Humulene, Docosane, Octadecane, Eicosane, Germacrene-D, Nonacosane in *C. arvense* [85]; Thymol in inflorescences of *C. pannonicum,* and *C. decussatum* were detected [65].

## 9. Pharmacological Studies

*Cirsium* has abundant pharmacological activities due to the presence of a wide range of phytochemicals. Till date, a broad spectrum of biological properties such as antimicrobial, antioxidant, analgesic, anticancer, hepatoprotective, and anti-inflammatory have been reported from the different species of *Cirsium* (Table 1).

### 9.1. Antimicrobial Activity

A huge number of studies have been carried out on various species of *Cirsium*, which proves that it possesses a great number of antimicrobial properties (Table 3). Nazaruk and Jakoniuk [50] used the flowers and leaves of *Cirsium rivulare* to test the microbicidal potential against bacterial strains namely *Klebsiella pneumoniae*, *Bacillus subtilis*, *Pseudomonas aeruginosa*, *Escherichia coli*, *Micrococcus* and *Staphylococcus aureus* and fungus *Candida albicans*. It was found that all *Cirsium rivulare* extracts exerted antiproliferative and bactericidal activity. However, water extract from leaves of *Cirsium* was observed to be more active against Gram-positive bacteria [50]. The antibacterial activity of five compounds, namely Tracin, 9,12,15-Octadecatrienoic acid, Luteolin, Hispidulin, and α-Tocopherol isolated from the *Cirsium arvense*, was tested against different bacterial strains. It was observed that Tracin, Luteolin, and Hispidulin exhibited marked protective efficacy against bacterial strains, whereas α-tocopherol showed moderate antibacterial activity. However, 9,12,15-Octadecatrienoic acid showed low bactericidal activity. In addition to it, the antifungal activity of these compounds was also monitored against six pathogenic fungi, *Trichophyton longifusus*, *Microsporum canis*, *Fusarium solani*, *Candida glabrata*, *Candida albicans*, and *Aspergillus flavus*. It was shown that all the phytocompounds had a moderate to low antifungal efficacy, except 9,12,15-Octadecatrienoic acid, which has no antifungal impact [35]. In another study, flowers and leaf extracts from *C. vulgare*, *C. rivulare*, *C. palustre*, *C. oleraceum*, or *C. arvense* have been shown to exhibit a significant antimicrobial effect against *P. aeruginosa*, *S. aureus*, and *B. subtilis*. Flower extract of *Cirsium* species showed a more potent antimicrobial effect as compared to the leaf extract [87]. Moreover, flavonoids present in *C. oleraceum* also displayed substantial antibacterial and antifungal effects [88]. Nazaruk et al. [89] investigated the antibacterial effects of *Cirsium* species against bacterial strains such as *S. aureus*, *P. aeruginosa*, *B. subtilis*, and the fungal strain *C. albicans*. Among the *Cirsium* species studied, *C. palustre* had the strongest antibacterial activity [39]. However, the protective potential of bioactive components extracted from *C. canum* was determined against the Gram-negative and positive bacteria. It was noticed that the extract and fractions of *C. canum* showed no effect on the growth of Gram negative bacteria. However, it showed potent inhibitory activity against the microbes *S. pneumonia*, *S. aureus*, *S. epidermidis*, *B. subtilis*, and *B. cereus* [63]. Moreover, Strawa et al. [90] found that hexane and defatted methanol extracts of *Cirsium* roots have a strong bactericidal effect against Gram-positive and Gram-negative bacteria with MIC and MBC values ranging from 25 to 200 μg/mL. In another investigation, the microbicidal effect of crude extract and fractions of *C. scabrum* was determined against twenty-two Gram (+) and thirteen Gram (−) microbial strains. Ethyl acetate and Butanol fraction of *C. scabrum* showed profound effect against the *S. aureus* and *D. hominis* strains [47]. Flavonoids isolated from *C. japonicum* demonstrated mild antibacterial activity against two strains of human skin bacilli and six strains of *S. aureus* [10]. Ciryneol D, isolated from the *C. setosum*, inhibited the development of mycelium in a variety of fungi [91].

**Table 3.** Antimicrobial and antioxidant potential of different *Cirsium* species.

| S. No | *Cirsium* Species | Application | Model | Detailed Information | References |
|---|---|---|---|---|---|
| 1. | *C. scabrum* | In vitro | *S. aureus*, *Dermabacter hominis* | Moderate activity | [47] |
| 2. | *C. canum* | In vitro | Gram-positive Bacteria | Inhibitory activity against *S. aureus* and *S. pneumoniae* | [63] |
| 3. | *C. arvense* | In vitro | *S. aureus, S. typhi* | Zone of inhibition: 9–32 mm | [35] |
| 4. | *C. oleraceum* *C. palustre* *C. rivulare* *C. vulgare* *C. arvense* | In vitro | *S. aureus* *P. aeruginosa* *B. subtilis* *C. albicans* *Micrococcus luteus* *E. coli* | Minimum inhibitory concentration range from 3.12–50 mg/mL | [50,89] |
| 5. | *C. hypoleucum* | In vitro | *S. aureus* | Inhibitory activity against *S. aureus* at 32 µg/mL | [92] |
| 6. | *C. setidens* | In vitro | - | DPPH activity: $IC_{50}$ value of 45.14 g/mL | [93] |
| 7. | *C. japonicum* | In vitro | Neuronal cells | More levels of heme oxygenase, thioredoxin reductase, antioxidative enzymes | [94] |
| 8. | *C. arvense* | In vitro | - | DPPH activity:118 µg/mL | [46,52] |
| 9. | *C. palustre* | In vitro | - | CAF(Flower) > CAR (Root) > CAL (Leaf) > CAS (Stem) | [56] |
| 10. | *C. leucopsis* *C. sipyleum* *C. eriophorum* | In vitro | - | DPPH inhibition: 4–38.98 % | [95] |
| 11. | *C. oleraceum* *C. rivulare* | In vitro | - | ABTS scavenging activity: >85% | [96] |
| 12. | *C. setidens* | In vivo | Wister albino rats | DPPH inhibition: 2.15–30% | [21] |
| 13. | *C. arvense* *C. oleraceum* *C. palustre* *C. rivulare* | In vitro | - | Total antioxidant activity: 0.98 to 2.71 mM/L | [39] |

*9.2. Antioxidant Activity*

Various *in vitro* investigations indicated that the roots, leaves, and flowers of *C. arvense* had a robust antioxidant effect (Table 3) [46]. Hossain et al. (2016) examined ethanolic extract of *C. arvense* for antioxidant activity. It was observed that ethanolic extract of *C. arvense* showed marked antioxidant activity [52]. The antioxidant potential of various extracts from *C. vulgare*, *C. rivulare*, *C. palustre*, *C. oleraceum*, and *C. arvense* was studied by Nazaruk [39], and the total antioxidant activity was found to be in the range of 0.98 to 2.71 m/mL [39]. In another investigation, crude aqueous extracts of the *Cirsium* species were monitored for total antioxidant potential through the ABTS technique. It was observed that *C. vulgare* (2.31 m/mL), *C. rivulare* (2.78 m/mL), *C. palustre* (2.78 m/mL), *C. oleraceum* (2.76 m/mL), and *C. arvense* (2.74 m/mL) showed the remarkable antioxidant activity [89]. In another study, Lee et al. [89] used the DPPH free radical test to determine the antioxidant capacity of leaves and root extracts of *C. setidens*. The $IC_{50}$ values for the butanolic fraction of leaves and roots were observed to be 33.53 g/mL and 9.75 g/mL, respectively. The free radical scavenging potential of *C. setidens* was found to be more than the tocopherol

and ascorbic acid [89]. Moreover, it was observed that the methanolic extract of *C. rivulare* roots exhibited more DPPH scavenging activity in contrast to the hexane extract [90]. Balanophonin, Vanillic acid, and Kaempferol-3-*O*-L-rhamnopyranoside purified from the *C. sipyleum*, *C. eriophorum*, and *C. leucopsis* showed huge antioxidant activity [95]. In another study, it was found that hexane extracts of *Cirsium* species contain the active component Linoleic acid, which displayed a remarkable antioxidant activity [90]. Chlorogenic acid was identified as the potent antioxidant compound among the different species of *Cirsium* in Poland [90]. In another investigation, the major antioxidant components in *Cirsium japonicum* were found to be Luteolin and Silibinin [97]. Moreover, Luteolin, apigenin, and their glucosides present in different species of *Cirsium* also showed antioxidant and hepatoprotective effects [98]. In addition to it, major flavonoids Pectolinarin and Pectolinarigenin present in *Cirsium* exhibited strong antioxidant activity [58]. In another study, Apigenin, Diosmetin, and Silicristin were found to be present in significant amounts in different species of *Cirsium* and also displayed potent DPPH radical-scavenging properties [99].

### *9.3. Antiproliferative Activity*

The antiproliferative potential of methanolic and chloroform extracts from different parts of *C. arvense* was tested against C6 cells (Rat brain tumor cells), Hela cells (Mammalian uterine carcinoma), and Vero cells (African green monkey renal cells). The root extracts of *C. arvense* revealed maximum inhibitory activity against the proliferation of C6 cells. However, the stem extracts showed huge inhibition against the Vero and HeLa cell lines. Moreover, the phytoconstituents isolated from the *C. arvense* showed antiproliferative activity in the order: Arvense 1 < Stigmasterol < Linarin < Daucosterin < 5-FU [46]. In addition to it, Tocospiro C, Tocospiro A, and Tocospiro B isolated from the leaves and stem of *C. setosum* showed inhibitory action against mammalian stomach cancer, ovarian cancer, lung adenocarcinoma (A549), hepatoma (Bel7402), and colon cancer (HCT-8) cells. Tocospiro C and B exerted maximum selective inhibition against the mammalian colon cancer cells, and $IC_{50}$ was observed to be 0.03 µM and 0.12 µM, respectively (Table 4). However, Tocospiro A revealed very low inhibitory activity against HCT-8 cells, and $IC_{50}$ was found to be 12.8 µM. All the compounds showed similar inhibition against the other cancer cell lines and displayed $IC_{50} > 20$ µM [70]. In another study, it was observed that aerial parts of *C. setidens* contain major phytoconstituents Tocopherol, 24-Hydroperoxycycloart-25-en-3β-ol, trans-Phytol, 9, 12, 15-Octadecatrienoic acid, Hexadecanoic acid, and Sitosterol. It was observed that maximum cytotoxicity against the mammalian cancer cell lines was exhibited by the 24-Hydroperoxycycloart-25-en-3β-ol, and $ED_{50}$ was observed to be in the range from 2.66 to 11.25 µM. The remaining phytoconstituents showed negligible cytotoxic action on the cancer cell lines [23]. Similarly, the cytotoxic potential of leaf extracts of *C. scabrum* was monitored against the mammalian macrophage cell line. It has been noticed that methanolic, and petroleum ether fractions of *C. scabrum* displayed selective cytotoxic activity with the $IC_{50}$ value of 11.53 and 12.12 µg/mL, respectively [47].

In another investigation, the antiproliferative activity of essential oil purified from *C. palustre* and *C. rivulare* was determined against the adenocarcinoma cell line. It revealed moderate inhibitory action against the adenocarcinoma with an $IC_{50}$ value of 110–140 g/mL [73]. Likewise, extracts of *C. palustre* and *C. arvense* showed a little cytotoxic effect on the normal skin fibroblasts in a dose-dependent manner [87].

**Table 4.** Anticancer and anti-inflammatory activities of *Cirsium* species.

| S. No | *Cirsium* Species | Application | Model | Detailed Information | References |
|---|---|---|---|---|---|
| 1. | *C. scabrum* | In vitro | J774 cancerous cell line | $IC_{50} = 11.53$ μg/mL | [47] |
| 2. | *C. rivulare* | In vitro | MCF-7 and MDA-MBA-breast cancer cell line | $IC_{50} = 110$ to $140$ μg/mL | [73] |
| 3. | *C. setosum* | In vitro | HCT8 colon cancer cells | $IC_{50} = 0.03$ μM | [70] |
| 4. | *C. tenoreanum* | In vitro | MCF7 breast cancer cells | 73% cell death | [24] |
| 5. | *C. arvense* | In vitro | HeLa and C6 cell lines | CAR > CAF > CAL | [46] |
| 6. | *C. setidens* | In vitro | Lung, skin, ovarian, and colon cancer cells | $ED_{50} = 2.66$ to $11.25$ μM | [23] |
| 7. | *C. japonicum* | In vitro | Breast cancer cells | Reduction in angiogenesis by lowering the production of VEGF, Akt, and ERK in MDA-MB-231 cells | [100] |
| 8. | *C. japonicum* | In vitro | MCF-7 cells | Arresting the cell cycle in the G1 phase and induced apoptosis | [101] |
| 9. | *C. chanroenicum* | In vitro | RAW macrophage cells and murine leukemia cells | Inhibition of cyclooxygenase and leukotriene production | [59] |
| 10. | *C. subcoriaceum* | In vivo | Murine model | $ED_{30} = 25$ mg/kg | [58] |
| 11. | *C. japonicum* | In vitro | Macrophage cell line Mast cell line | Reduction in pro-inflammatory cytokines, NO and NF-κB in Mast Cells | [102,103] |

*9.4. Analgesic and Anti-Inflammatory Activity*

The water extracts from the aerial portion of *C. subcoriaceum* and its active component pectolinarin were monitored for analgesic and anti-inflammatory activity (Table 4). It was observed that crude extract of *C. subcoriaceum* and pectolinarin hampered the acid-induced writhing in mice in a concentration dependent manner. The pectolinarin ($ED_{50}$ 28.44 mg/kg) was observed to be more effective as an analgesic as compared to the crude extract of *C. subcoriaceum* ($ED_{50}$ 83.18 mg/kg). It has been observed that Pectolinarin showed similar protective efficacy as a standard analgesic compound, Acetylsalicylic, at a similar concentration. The water extract of *C. subcoriaceum* and pectolinarin hindered the edema induced by carrageenan. The $ED_{30}$ of *C. subcoriaceum* and pectolinarin was detected to be 25 mg/kg and 3.7 mg/kg, respectively [58]. Similarly, Pectolinarigenin and Pectolinarin extracted from the aerial parts of *C. chanroenicum* hindered cyclooxygenase-2 mediated prostaglandin E2 synthesis and leukotrienes in LPS-treated macrophages, which resulted in lesser production of eicosanoid. Moreover, oral administration of Pectolinarigenin and Pectolinarin declined inflammation and allergy in the animal control group. Thus, these components might play some part in the anti-inflammatory properties of crude extract of *C. chanroenicum*. COX inhibitors are utilized as anti-inflammatory drugs, while 5-lipoxygenase inhibitors have anti-allergic action [59]. In another research, the crude extract and its flavonoid Cirsimaritin caused the marked inhibition of nitric oxide and nitric oxide synthase expression in macrophage cells. It repressed the production of tumor necrosis factor-α, interleukin-6, and NO in macrophage cells induced by lipopolysaccharide. Moreover, pretreatment with Cirsimaritin caused the reduction of phosphorylation of IκBα and Akt in LPS-induced macrophage cells. Cirsimaritin decreased the induction of FOS and STAT3 (signal transducer and activator of transcription 3) signaling in macrophages which

depicts its anti-inflammatory nature [102]. Likewise, Silibinin purified from *C. japonicum* decreased the growth of human mast cells and caused a reduction in the expression of pro-inflammatory cytokines. Furthermore, Silibinin declined the phosphorylation of IκBα and NF-κB transcriptional activity in stimulated mast cells. Therefore, it can be employed for the cure of mast cells mediated allergic inflammatory ailments [103].

### 9.5. Hepatoprotective Activity

The hepatoprotective activity of water extract of roots and leaves of *C. arisanense* was monitored in mammalian hepatocellular carcinoma cell lines and mice (Table 5). Roots and leaves of *C. arisanense* shielded the hepatocellular carcinoma cells against tacrine-stimulated hepatotoxicity and decreased the expression of hepatitis B surface antigen. Furthermore, roots and leaves of *C. arisanense* ameliorated the hepatic damage in mice induced due to tacrine as they lowered the concentration of serum glutamate-pyruvate transaminase (SGPT) and serum glutamic oxaloacetic transaminase (SGOT). These effects of the roots of *C. arisanense* might be due to an enhancement in the concentration of liver glutathione [42]. Likewise, the administration of butanol extract of *C. setidens* at 500 mg/kg caused a reduction in the hepatic damage in rats induced due to $CCl_4$. Treatment of *C. setidens* elevated the concentration of antioxidant markers glutathione peroxidase, glutathione peroxidase, and superoxide dismutase (SOD) in the liver of rats. Histological studies substantiated the biochemical analysis, indicating that the extract of *C. setidens* caused a remarkable decrease in hepatic ballooning degeneration [21]. Moreover, the active components Pectolinarin and Pectolinarigenin present in methanolic extracts of *C. setidens* leaves decreased the hepatic derangement in rats induced due to galactosamine toxicity. The concentration of SGOT, SGPT, alkaline phosphatase (ALP), and lactate dehydrogenase (LDH) were observed to be decreased after the treatment of Pectolinarin and Pectolinarigenin, indicating the hepatoprotective potential of both of the components. Treatment with Pectolinarin and Pectolinarigenin boosted the antioxidant enzymes glutathione, glutathione transferase, glutamylcysteine synthetase, glutathione reductase, and SOD. It revealed that the hepatoprotective activity of Pectolinarin and Pectolinarigenin is attributed to the induction of the antioxidant system [60]. In another investigation, the crude extracts of *Cirsium* normalized the levels of SGOT and SGPT in mice that had been raised by $CCl_4$ injection. *Cirsium japonicum* and *Cirsii herba* decreased the $CCl_4$ stimulated liver necrosis and restored the levels of hepatic antioxidant enzymes and malondialdehyde [104].

**Table 5.** Other pharmacological activities of *Cirsium* species.

| Activity | *Cirsium Species* | Application | Model | Detailed Information | References |
|---|---|---|---|---|---|
| Oviposition stimulatory | *C. japonicum* | In vitro | *Ostrinia zealis* | Extract potently induced oviposition by females | [41] |
| Allelopathy | *C. creticum* | In vivo | Radish Lettuce Cress | Inhibitory activity on germination | [17] |
| Enzyme inhibition activity | *C. japonicum* | In vitro and In vivo | Chondrocytes | Decrease the levels of Hif-2α, metalloproteinases, and cyclooxygenases | [104] |
| | Flavonoids of *C. japonicum* | In vitro | Aldose reductase inhibitor | $IC_{50}$ values of 0.21 µg/mL and 0.77 µM | [54] |
| | *C. leucopsis* *C. sipyleum* *C. eriophorum* | In vitro | Acetyl-and butyryl-cholinesterase inhibitory activity | 16–57% Inhibition | [105,106] |
| | *C. japonicum* | In vivo | Murine model | Reduction in the levels of lipoprotein lipase and fatty acid synthetase | [106] |

**Table 5.** *Cont.*

| Activity | *Cirsium* Species | Application | Model | Detailed Information | References |
|---|---|---|---|---|---|
| Hepatoprotective | *C. japonicum* Cirsii herba | In vivo | C57BL/6 Mice | Decrease in liver necrosis restored the hepatic antioxidant enzymes and malondialdehyde | [107] |
| | *C. arisanense* | In vitro and In vivo | Hep 3B Cells and Mice | Reduction in Hepatitis B surface antigen. Declined the levels of SGOT and SGPT | [42] |
| | *C. setidens* | In vivo | Mice | Decrement in hepatic damage in rats induced due to $CCl_4$ and hepatic ballooning degeneration | [21,83] |
| Nephroprotective | *C. japonicum* | In vivo | Murine Model | Decrease the levels of Cholesterol and triglycerides | [108] |
| | *C. japonicum* | In vitro | 3T3-L1 Cells | Enhancement in insulin-stimulated glucose uptake | [109] |
| Immunomodulatory | *C. japonicum* | In vivo | Murine model | Induction of humoral and cellular immune responses. Activation of complement pathway and Natural killer cell activity | [57] |

### 9.6. Immunomodulatory Activity

*C. japonicum* and its phytoconstituent Pectolinarin hindered the growth of tumors in mice and enhanced the humoral and cellular immune responses (Table 5). It boosted the complement pathway in the tumor-bearing mice and also caused the improvement in the transformation of spleen cells as well as natural killer cell activity [57].

### 9.7. Anticancer Activity

There are numerous studies indicating the anticancerous nature of *Cirsium* (Table 4) and its promising therapeutic ability to prevent malignancies. The Pectolinarin and 5,7-Dihydroxy-6,4-dimethoxyflavone obtained from *C. japonicum* were monitored for anticancer activity in the mouse. It was observed that both of the compounds exhibited a remarkable reduction in the multiplication of tumor cells in a concentration-dependent manner [53]. Similarly, the extract of *Cirsium japonicum* and Cirsimaritin caused a reduction in the breast cancer cells indicated by the suppression of expression of Akt, VEGF, and ERK in MDA-MB-231 cells. Furthermore, *C. japonicum* extract exhibited an antiproliferative effect in MCF-7 cancer cells by restricting the cell cycle in the G1 phase and also triggered cell death by influencing mitochondrial apoptotic pathways [100]. Triterpenes present in *Cirsium setosum* revealed moderate cytotoxicity against human colon and ovarian cancer cells. [101], while 3β-Hydroxy-22-oxo-20-dandelion-30-oleic acid displayed a potent selective inhibition on the ovarian tumor cell line A2780 [110]. Silybin, derived from *Cirsium japonicum*, hindered gastric cancer cells by reducing the production of cell cycle proteins and blocking gastric tumor cells in the G2/M phase, which ultimately resulted in apoptosis (Figure 12) [111].

### 9.8. Oviposition Stimulatory Activity

Treatment of roots essential oil of *C. japonicum* induced the oviposition in *Ostrinia zealis* (Table 5). It was observed that root essential oil of plants contains majorly Aplotaxene, which might serve as a stimulator of oviposition [41]. In another investigation, treatment of *C. japonicum* extract to ovariectomized rats caused a marked decline in cholesterol, body weight, and triglyceride, as well as remarkable enhancement in estradiol and bone mineral density. Moreover, Molecular docking studies revealed that the active phytoconstituents of *C. japonicum* have a binding affinity with the ligand-binding sites of the estrogen receptor. It indicated the potential of *C. japonicum* extract in the relief of symptoms of pre-menopause and post-menopause [100].

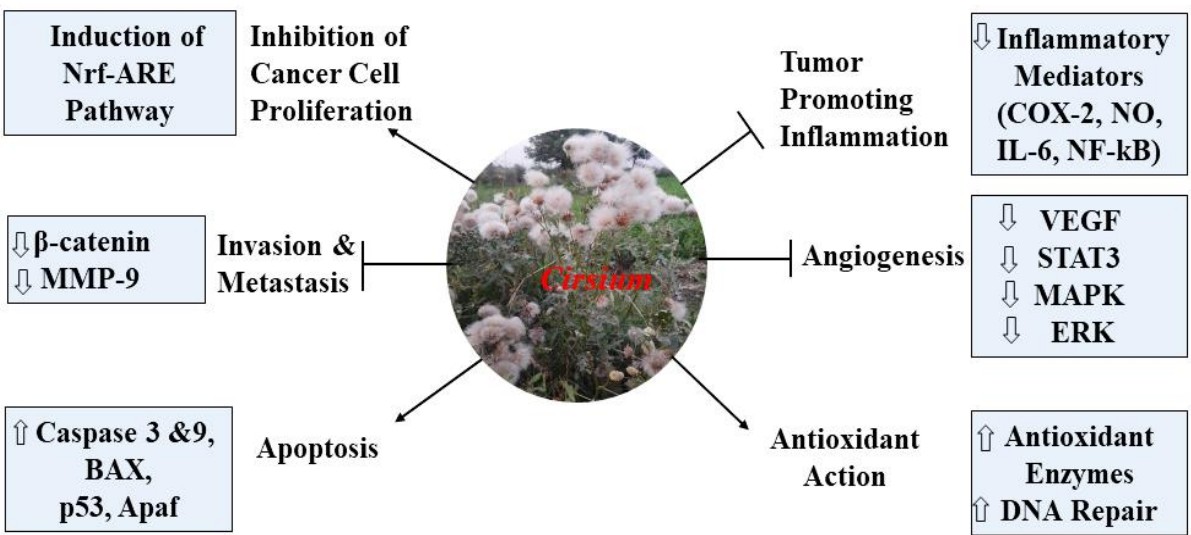

**Figure 12.** Mode of action of Silybin against gastric cancer.

*9.9. Anti-Anxiety Effect*

The ethanolic extract of *Cirsium japonicum* was screened for the anti-anxiety effect in mice. The extract of *C. japonicum* showed no effect on the movement of mice in the open-field test. However, it increased the exploration in the undefended center zone. Administration of plant extract also enhanced the duration of mice in the elevated plus-maze test, which pointed out that the *Cirsium japonicum* exhibited anti-anxiety properties. It was observed that *C. japonicum's* anti-anxiety effects were equivalent to those of the benzodiazepine. Further, the anti-anxiety effect of *C. japonicum* was confirmed by examining its effect on human neuroblastoma cells. Treatment of *C. japonicum* upregulated the influx of chloride ions in neuroblastoma cells in a concentration-dependent manner, which was reduced by coadministration of bicuculline [112]. In another investigation, *C. japonicum* had an antidepressant-like impact on mice in the forced-swimming test by dramatically lowering immobility. It was observed that administration of *C. japonicum* did not boost the locomotor activity in the open field test. Moreover, it enhanced the influx of Cl- ions without affecting the uptake of monoamine in mammalian neuroblastoma cells. Only Luteolin, one of the active components of the *C. japonicum* extract, revealed similar antidepressant-like effects. Therefore, the antidepressant-like action of *C. japonicum* extract was believed to be due to the presence of luteolin in it [113].

*9.10. Nephroprotective*

*Cirsium* is nephroprotective in nature, as it shields the kidney from various toxins (Table 5). Pectolinarin and flavones purified from *Cirsium japonicum* were studied for their nephroprotective activity in diabetic rats induced by streptozotocin followed by a high-carbohydrate/high-fat diet. Both flavones exhibited antidiabetic activity in rats. However, a combination of Pectolinarin and 5,7-Dihydroxy-6,4'-dimethoxy flavone in diabetic rats was found to be more potent in the reduction of Cholesterol, Glucose, and Triglycerides. Treatment of flavones in diabetic rats caused the reversal of abnormal levels of glucose metabolism-related enzymes. It increased the concentration of adiponectin in diabetic rats, but there was no discernible impact of the flavones on the abnormal concentration of insulin or leptin and glucose transporter 4 [108]. Moreover, *Cirsium japonicum* flavones improved adipocyte development by enhancing the levels of PPARγ. It enhanced the insulin-induced glucose intake in adipocytes, which is probably due to the more levels of adiponectin and GLUT4 [109]. In another study, the glucosidase enzyme is inhibited by taraxastane-type triterpenoids derived from *Cirsium setosum* [110]. Additionally, the leaves of *Cirsium maackii* and its flavonoids showed an antidiabetic effect by hampering the production of glycation end products [114].

### 9.11. Other Therapeutic Effects

Treatment with *C. japonicum* (50–100 mg/kg/day) enhanced cognitive skills by diminishing the oxidative stress in amyloid β-peptide-induced mice, and it can be utilized as a potent agent for the cure of Alzheimer's disease (Figure 13). In addition to it, many investigators have examined the hemostatic action and mechanism of different *Cirsium* species. Administration of *Cirsium setosum* extract exhibited a substantial activity on hemostasis, blood coagulation, and hemorrhage in rats [115,116]. Wang (2018) also reported the hemostatic action of nano-scale constituents in *Cirsium charcoal* [117]. Moreover, Shikokiol A obtained from *C. nipponicum* roots was tested in vitro for enzyme inhibitory activities. It was observed to be a potent inhibitor of the non-heme iron-containing enzyme, i.e., lipoxygenase, in guinea pig tracheal contraction [118,119].

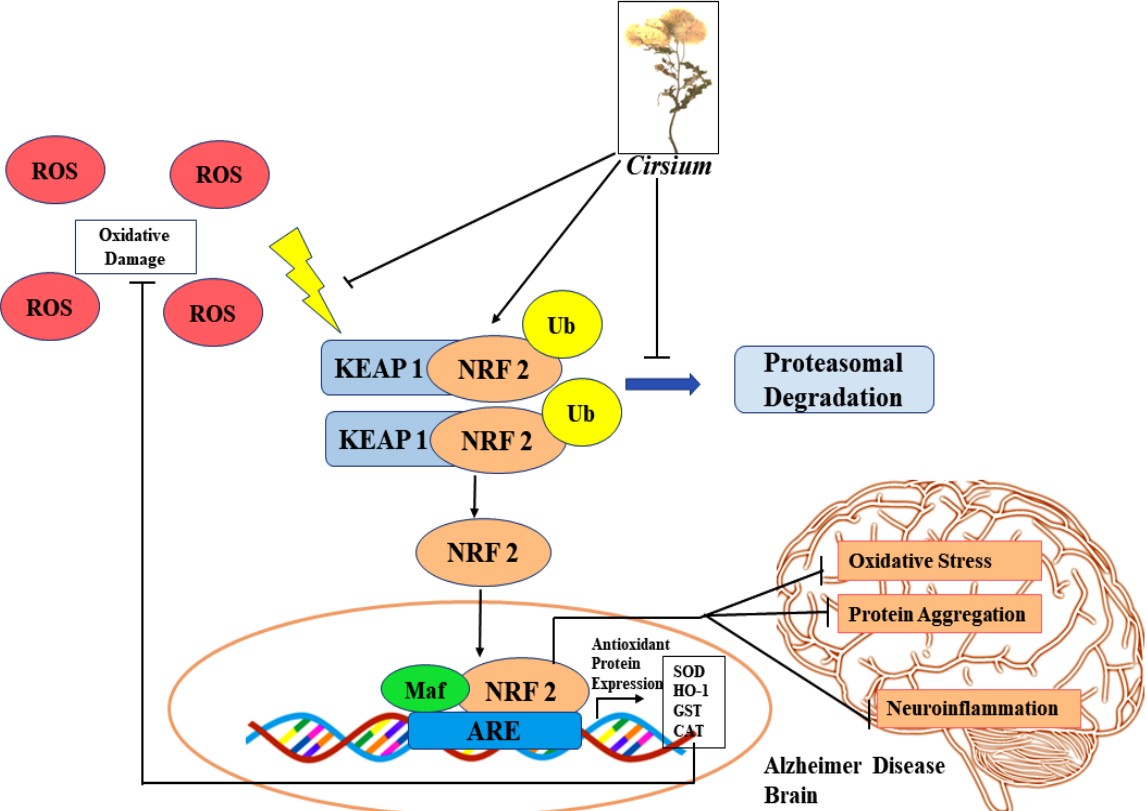

**Figure 13.** Mode of action of *Cirsium* species against Alzheimer's disease.

The *C. sipyleum*, *C. leucopsis,* and *C. eriophorum* exhibited 16–57% inhibition against acetyl- and butyryl-cholinesterase activity [21,105]. Similarly, Lee et al. [54] showed that ethanolic fraction and flavonoids of *C. japonicum* displayed aldose reductase inhibition activity with $IC_{50}$ values of 0.21 µg/mL and 0.77 µM, respectively [54]. In another study, *Cirsium japonicum* and its constituent apigenin caused a marked reduction in the expression of Hif-2α and decreased the levels of metalloproteinases and cyclooxygenases in chondrocytes. This study depicted the potential of *Cirsium japonicum* and its constituents for the development of therapeutics for hindering osteoarthritis [104]. The extracts of *C. japonicum* hindered the adipogenesis in adipocytes by diminishing the concentration of triglycerol. The chloroform fraction of *C. japonicum* was observed to reveal the maximum inhibition of adipocyte differentiation. The extract downregulated the expression of lipoprotein lipase, PPARγ, adiponectin, and fatty acid synthetase intricated in adipogenesis. Therefore, *C. japonicum* extract can be used as an ideal candidate for the treatment of obesity [120].

## 10. Herbal Formulations and Clinical Trials

MS-10, a formulation of *Cirsium* and Thyme extract, was examined on 71 premenopausal women for 28 and 84 days in a randomized, double-blind clinical trial. Treatment of MS-10 significantly reduced the onset of menopause by 48%. Moreover, MS-10 enhanced the insulin-like growth factor-1 (IGF-1) and estrogen, which might show protective inhibitory effects on menopause and aging in women. MS-10 also promoted bone health in women by increasing bone formation and absorption indicators such as osteocalcin, alkaline phosphatase, collagen, and N-telopeptides of type I collagen. It was observed that MS-10 therapy improved the levels of cortisol and upgraded the psychological well-being index in women [121]. Another herbal formulation, Fufang Zhenzhu Tiaozhi (FTZ), contains *Ligustrum lucidum*, *Citrus medica*, *Coptis chinensis*, *Atractylodes macrocephala*, *Panax*, *Cirsium japonicum*, *Salvia miltiorrhiza*, and *Eucommia ulmoides* [122]. FTZ has been employed clinically for the cure of hyperlipidemia, diabetes, osteoporosis, and atherosclerosis [123,124]. Zhang et al. [125] reported that the FTZ treatment lessened cardiac hypertrophy in mice through the downregulation of expression of miR-214 and upregulation of SIRT3 expression. Several studies indicated the therapeutic potential of FTZ is linked to a variety of pharmacotherapeutic activities [126,127]. Moreover, various human and animal studies demonstrated that FTZ has an excellent ability to reduce total cholesterol, triglyceride in the blood. This formulation also exhibited protective action against metabolic diseases such as atherosclerosis, hyperlipidemia, and hepatic infections by modifying the concentration of glucose and lipids in the blood [128,129]. Diao et al. (129) demonstrated that FTZ decreased atherosclerosis by hindering endothelial–mesenchymal transition through the β-catenin pathway. Treatment of FTZ bettered dyslipidemia and dysfunctioning of endothelial cells in the atherosclerotic mice. Moreover, FTZ administration caused the reduction of total bad cholesterol and boosted HDL. It also enhanced the levels of endothelial markers such as CD31 and cadherin and diminished the mesenchymal markers, signifying that it impeded the endothelial–mesenchymal transition.

In another study, the protective efficacy of FTZ was determined in a mouse model of Polycystic ovary syndrome (PCOS). It was found that FTZ remarkably increased the levels of adiponectin, thus modifying adipose-ovary crosstalk to decline PCOS. Moreover, the administration of FTZ decreased the disruption of the estrous cycle, cystic follicles, and insulin resistance [96]. In another investigation, Yang et al. [130] described that FTZ repressed renal inflammation and fibrosis by the suppression of NF-κB and IL-17. Administration of FTZ also reduced the levels of urea, glucose, triglycerides, cholesterol, fibronectin, and collagen. Similarly, a multicenter, randomized, double-blind trial also reported the protective potential of FTZ on diabetic coronary heart disease [131]. In another clinical study, flower extract of *Cirsium japonicum* improved the wrinkles and elasticity of the skin, and it can be preferred as an active component of antiaging cosmetics [132].

## 11. Patents

A formulation comprising *Cirsium japonicum* extract as a potent agent for the induction of melanogenesis: The melanogenesis-stimulating formulation includes *Cirsium japonicum* extract as an active component, and this composition can be safely utilized for the prevention and treatment of vitiligo, white hair, or hypopigmentation [133].

A process for the cure of fatty liver: A method was designed for the preparation of hepatoprotective formulation from the extract of *Cirsium* (Table 6) [134].

A method for the enhancement of Lipolysis: The extracts of *Cirsium* showed huge lipolysis when orally administrated or dermatologically apply through local administration. It pointed out the potential of *Cirsium* in the control and treatment of obesity [135].

Anti-acne formulation: A composition was prepared from *Quercus robur*, *Sesamum indicum*, *Houttuynia cordata*, *Cirsium japonicum*, and *Thuja orientalis* as potent agents for the cure of acne. This composition revealed maximum antibacterial activity against *C. acnes* [136].

**Table 6.** Details of patents of *Cirsium*.

| S. No | Title of Patent | Applicant | Published Application Number |
|---|---|---|---|
| 1. | Composition containing *Cirsium japonicum* extract as active ingredient for stimulating melanogenesis | Biospectrum Inc. | US20210361559A1 |
| 2. | Immunoregulatory composition containing *Cirsium maritimum* extract | Kochi Prefectural University Corp Kochi | JP6882730B2 |
| 3. | Organic extract of plant of genus *Cirsium*, and application and composition thereof | Zhejiang Wolwo Bio Pharmaceutical Co., Ltd. | CN112022892A |
| 4. | Method for treating fatty liver | NPO Amami Functional Foods Study Group University of the Ryukyus Amino UP Co., Ltd. | US10653740B2 |
| 5. | Ceramide production enhancer and moisturizer | Kao Corp | US9682029B2 |
| 6. | Composition for preventing, ameliorating, or treating acne symptoms using natural extracts as active ingredients | Celim Biotech Co., Ltd. | US11154580B1 |
| 7. | Preparation composition for external use for skin and bath agent composition | Kao Corp | JPH09208483A |
| 8. | Composition for bubble bath | Kao Corp | JPH10147516A |
| 9. | Adiponectin secretion promoting agent | NPO Amami Functional Foods Study Group Tokunoshima Town Osaka University NUC | US20210283207A1 |
| 10. | Lipolysis acceleration method | Kao Corp | US5698199A |
| 11. | Fat accumulation inhibitor, drug, prophylactic or therapeutic agent for fatty liver, food or drink, and method for producing fat accumulation inhibitor | NPO Amami Functional Foods Study Group Amino UP Chemical Co., Ltd. University of the Ryukyus | US20160184378A1 |

Enhancer and moisturizer of Ceramide production: Composition containing *Cirsium japonicum* singly or in combination with *Chenopodium hybridum*, *Melia toosendan*, *Indigofera tinctoria*, *Catalpa ovata*, and *Tagetes erecta* showed an excellent capacity for the enhancement of ceramide [137].

## 12. Pharmacokinetics

Pharmacokinetics serves a critical role in preclinical drug development, screening of toxicity of the drug, and optimization of the concentration of the drug. It is measured as an effective way of detecting the potential active constituents and explaining the mode of action of plants or drug formulations. So far, there are very few scientific investigations in the literature on the pharmacokinetic behavior of *Cirsium*. A liquid chromatography–mass spectrometry (LC-MS) technique was employed to monitor the different flavonoids of *C. setosum* in rat plasma. It was observed that Rutin, Acacetin, Naringin, Wogonin, and

Quercetin were the long-acting constituents of the *C. setosum*, with more elimination time and bioavailability [138].

Similarly, Zhang et al. [139] detected twenty-seven flavonoids in the blood, bile, and urine of rats after the administration of *Cirsium japonicum* through the UPLC-MS. In another study, after treatment with the extract of *Cirsium japonicum* in rats, the maximum concentration of Linarin, Pectolinarigenin, Hispidulin, Pectolinarin, Diosmetin, Acacetin, and Apigenin in plasma was observed to be 86 ng/mL, 6 ng/mL, 32 ng/mL, 876 ng/mL, 37 ng/mL, 19 ng/mL and 148 ng/mL, respectively. Pectolinarin, Linarin, Pectolinarigenin, Hispidulin, Diosmetin, and Acacetin were absorbed rapidly and reached their maximum concentration in plasma in five minutes (Table 7). However, Apigenin was absorbed slowly and reached its maximum concentration after 360 min of its administration [140].

**Table 7.** Pharmacokinetic information of *Cirsium in vivo*.

| Model | Administration Method | Quantitative Method | Details | References |
|---|---|---|---|---|
| Sprague-Dawley rats | Oral | UHPLC-Q-TOF-MS | Quercetin, Luteolin, Diosmetin, Cirsimarin, Linarin, Apigenin, Cirsimaritin, Pectolinarin, Tilianin, Hispedulin, Pectolinarigenin, Acacetin were detected | [139] |
| Sprague-Dawley rats | Oral | LC-MS | Maximum $C_{max}$ for quercetin = 513.2 ng/mL, while the minimum $C_{max}$ of diosmetin = 231.2 ng/mL $AUC_{0-t}$ value of Compounds (Higher Bioavailability) Quercetin = 6071 ng·h/mL Wogonin = 3789 ng·h/mL Naringin = 2808 ng·h/mL Acacetin = 2636 ng·h/mL Rutin = 1884 ng·h/mL $AUC_{0-t}$ value of Compounds (Lower Bioavailability) Diosmetin = 238.0 ng·h/mL | [138] |
| Sprague-Dawley rats | Oral | LC-MS/MS | $C_{max}$ of detected Compounds (ng/mL) Pectolinarin = 876 Diosmetin = 37 Pectolinarigenin = 6 Linarin = 86 Hispidulin = 32 Acacetin = 19 Apigenin = 148 $T_{max}$ of Pectolinarin, linarin, pectolinarigenin, hispidulin, diosmetin, acacetin = 5 min $T_{max}$ of Apigenin = 360 min | [140] |

## 13. Toxicology

A systemic safety study of *Cirsium* extract is required for the expansion of novel pharmaceuticals or drugs. Oral administration of *C. setidens* extract in rats for 28 days at the dose of 1.25, 2.5, and 5 g/kg body weight did not exhibit significant toxicological alterations such as mortality, hematology, and biochemical parameters. Treatment of *C. setidens* extract revealed normal histological architecture of the liver, heart, spleen, and kidney, which indicated the wide safety index of the plant extract [141]. Similarly, treatment with extract

of *C. japonicum* at the concentration of 2 g/kg body weight for 15 days revealed no marked toxic effects and mutagenicity [142].

## 14. Nanoformulations of *Cirsium*

Shin et al. [143] examined the *C. setidens*-derived selenium nanoformulations for their protective potential against oxidative stress. The nanoparticles of *C. setidens* extract unveiled zeta potential of −27.4 mV with a particle size of 117.8 nm. It was noticed that nanoformulation of *C. setidens* extract displayed more antioxidant and antibacterial activities as compared to the alone *C. setidens* extract. In addition to it, nanoformulation of *C. setidens* was observed to be harmless to normal fibroblast cell lines. However, they showed marked cytotoxic effects against A549 mammalian lung cancer cells through rupturing the mitochondrial membrane and nucleus [143]. Moreover, *C. arvense*-derived silver nanoparticles showed robust protective activity against *Escherichia coli* [144,145]. In another investigation, *Cirsium vulgare*-derived cobalt oxide nanoparticles augmented the electrocatalytic action for the monitoring of cysteine [146]. Moreover, *C. japonicum* extract served as a reducing and stabilizing agent for the synthesis of nontoxic silver nanoparticles. These silver nanoparticles showed 98% degradation of bromo phenyl blue in twelve minutes, which indicated the robust reductive potential of silver nanoparticles in the water cleansing and altering some organic harmful compounds to harmless components. In another investigation, *Cirsium arvense* derived copper nanoparticles (CA-CuNP) were examined for antibacterial and photocatalytic properties. CA-CuNP showed the photocatalytic potential and caused the complete degradation of Rhodamine B in thirty minutes. CA-CuNP exhibited microbicidal potential by hindering the growth rate of *S. aureus* and *E. coli* and displayed zone of inhibition 18 mm and 21 mm respectively [147,148]. In addition to it, water extract of *Cirsium setosum carbonisata* (CSC) was used for the synthesis of carbon dots (CD), which were spherical and even in size and unveiled a little noxiousness against macrophage cells. Moreover, tail and liver bleeding experiments showed a lower bleeding time in CSC-CD-treated mice in contrast to normal saline-treated mice. It was noticed that CSC-CD can enhance the extrinsic blood coagulation pathway and stimulate the fibrinogen proteins, which are projected towards the hemostatic effect of CSC-CD [149].

## 15. Conclusions and Future Perspectives

The *Cirsium* genus has been given a lot of attention due to its widespread usage in traditional medicine. The chemical compounds extracted from the *Cirsium* genus include mostly flavonoids, phenylpropanoids, and triterpenoids, which contribute to the multiple medicinal properties. From a phytochemical standpoint, a wide range of chemical structures have been identified from *Cirsium* species of varying distribution, which can result in a variety of biological effects. The current review suggests a wide range of potential applications of *Cirsium* in the field of pharmaceuticals, cosmetics, and health foods. Pharmacological studies revealed that *Cirsium* species have several biological activities, such as hepatoprotective, antibacterial, antioxidant, antitumor, and anti-inflammatory, which tends to untangle the various traditional applications of the *Cirsium* genus. The phytochemical studies of different *Cirsium* species and their renowned pharmacological activities could be exploited for pharmaceutic product development in the future. Furthermore, studies are required on less known *Cirsium* species, particularly on the elucidation of the mode of action of their biological activities. Further investigations should be carried out on pharmacodynamics, pharmacokinetics, and quality control system for *Cirsium*, which is critical for broadening its medicinal potential in the future. Moreover, the toxic and safe profile of *Cirsium* has not been well investigated; thus, further research is needed in this domain.

**Author Contributions:** G.A.: Formal analysis, writing—original draft, and data curation. G.K.: Writing—original draft. A.B.: Writing—original draft. H.S.S.: Chemical structures and software. G.B.: Methodology, software, and editing. G.A.N.: Writing—review and editing. V.M.: Writing—original draft—nanotechnology. A.S.: Conceptualization, methodology, supervision, validation, review, and editing. All authors have read and agreed to the published version of the manuscript.

**Funding:** This research received no external funding.

**Conflicts of Interest:** The authors declare no conflict of interest.

## Abbreviations

| | |
|---|---|
| ABTS | 2,2′-azino-bis(3-ethylbenzothiazoline-6-sulfonic acid) |
| ALT | Alanine aminotransferase |
| ARI | Aldose reductase inhibition |
| AST | Aspartate aminotransferase |
| Bel7402 | Human hepatocellular carcinoma |
| Cmax | Maximum plasma concentration |
| CCl4 | Carbon tetrachloride |
| COX-2 | Cyclooxygenase-2 |
| DPPH | 2,2-diphenyl-1-picryl-hydrazyl-hydrate |
| ED50 | Median effective dose |
| 5-FU | Fluorouracil |
| GABA | Gamma-aminobutyric acid |
| G1 | Growth 1 phase |
| G2/M | Growth 2 phase |
| HCT-8 | Human colon cancer cell line |
| HeLa cell | Henrietta Lacks cell |
| Hif-2$\alpha$ | Hypoxia-Inducible Factor-2$\alpha$ |
| HPLC-MS | High-performance liquid chromatography–mass spectrometry |
| IC50 | Half-maximal inhibitory concentration |
| IL-6 | Interleukin-6 |
| MBC | Minimum bacterial concentration |
| MCF-7 | Human breast cancer cell line |
| MDA-MB-231 | Human mammary carcinoma |
| MIC | Minimum inhibitory concentration |
| MMP3 | Matrix metalloproteinase-3 |
| MMP13 | Matrix metalloproteinase-13 |
| NF-$\kappa$B | Nuclear factor kappa B |
| NO | Nitric oxide |
| PPAR$\gamma$ | Peroxisome proliferator-activated receptor gamma |
| p-Akt | Protein kinase B |
| p-ERK | Extracellular signal-regulated kinase |
| SOD | Superoxide dismutase |
| VEGF | Vascular endothelial growth factor |
| CA-CuNP | Cirsium arvense-derived copper nanoparticles |
| CSC | Cirsium setosum Carbonisata |
| CD | Carbon dots |

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
