# Peer review of "Traditional Uses, Phytochemical Composition, Pharmacological Properties, and the Biodiscovery Potential of the Genus Cirsium"

_chemistry, doi:10.3390/chemistry4040079_

Round 1

Reviewer 1 Report

The topic of the thesis is not bad and it is interesting, but I have a major problem with the overall organization of the document.

I don't understand the layout of the tables, they are confusing. Information from the first table, where there is not a citation, is then simply copied into the others. The table, if it continues on the next page, must have a rewritten header. The individual information in the columns is scattered and it is not clear what belongs to what.

The formulas throughout the work are not uniform (different sizes, font spacing, bindings without indicating what is at the end - should it be H or CH3? - but that is shown on other bindings)

Once alpha is written with the symbol then as A.

The names of substances are not uniform - there are spelling errors and capital letters sometimes written in small letters. Try it in the name -O- (eg: Pectolinarigenin-7-O-Glucopyranoside) is written in italics -O- .

 The overall editing of the text is very bad. There are no spaces between paragraphs and headings. In some parts the text is scattered and so are the formulas. It should have a uniform order. Equal spacing formulas and labels.

 I recommend reworking the entire text on the graphic side, unifying the formulas and making an overall modification of the formatting. In this state, I would not recommend the text for publication.

Author Response

Comment 1: I don't understand the layout of the tables, they are confusing. Information from the first table, where there is not a citation, is then simply copied into the others. The table, if it continues on the next page, must have a rewritten header. The individual information in the columns is scattered and it is not clear what belongs to what.

Response: The layout of the tables has been changed and header also included when it is continuous on the second page. References also included in the first table.

Comment 2: The formulas throughout the work are not uniform (different sizes, font spacing, bindings without indicating what is at the end - should it be H or CH3? - but that is shown on other bindings)

Response:  The suggested changes has been done and maintained the uniformity throughout the manuscript.

Comment 3: Once alpha is written with the symbol then as A.

Response:  The suggested changes has been added.

Comment 4: The names of substances are not uniform - there are spelling errors and capital letters sometimes written in small letters. Try it in the name -O- (eg: Pectolinarigenin-7-O-Glucopyranoside) is written in italics -O- .

Response:  The suggested changes has been done and maintained the uniformity throughout the manuscript.

Comment 5: The overall editing of the text is very bad. There are no spaces between paragraphs and headings. In some parts the text is scattered and so are the formulas. It should have a uniform order. Equal spacing formulas and labels.

Response:  The required changes has been done.

Comment 6: I recommend reworking the entire text on the graphic side, unifying the formulas and making an overall modification of the formatting. In this state, I would not recommend the text for publication.

Response:  The suggested changes has been done.

Reviewer 2 Report

The study is well presented however there are some shortcomings which must be resolved

Line 40-42 should be cited. The following studies could be helpful.

DOI: 10.56042/ijtk.v21i3.31454, http://doi.org/10.36899/JAPS.2022.3.0484, https://doi.org/10.1016/j.chnaes.2021.03.009,  

The first paragraph must be about the significance of the medicinal and pharmacological importance of the plants.

Second paragraph could be about “Cirsium”   

Line 60-61 should be cited.

Add distribution and origin of the genus Cirsium.

Traditional uses should be area wise. Add details of the areas or regions about its uses.

Add significance and importance of the traditional uses at the start of the paragraph.

Conclusion should include future recommendations

Author Response

Comment 1: Line 40-42 should be cited. The following studies could be helpful.

DOI: 10.56042/ijtk.v21i3.31454, http://doi.org/10.36899/JAPS.2022.3.0484, https://doi.org/10.1016/j.chnaes.2021.03.009,  

Response:  The reference for Line 40-42 has been added.

Comment 2: The first paragraph must be about the significance of the medicinal and pharmacological importance of the plants.

Response:  The paragraph  about the significance of the medicinal and pharmacological importance of the plants has been added from the line number 44-49.

Comment 3: Second paragraph could be about “Cirsium”   

Response:  The suggested changes has been done.

Comment 4: Line 60-61 should be cited.

Response:  The reference for line 60-61 has been added.

Comment 6: Add distribution and origin of the genus Cirsium.

Response:  The distribution and origin of the genus Cirsium has been added from line number 94-97 and distribution also added in the table 1.

Comment 7: Traditional uses should be area wise. Add details of the areas or regions about its uses.

Response:  The  traditional uses regionwise added in the manuscript in line number 136 to 143.

Comment 8: Add significance and importance of the traditional uses at the start of the paragraph.

Response:  The importance of traditional uses at the start of the paragraph has been added from the line number 131-133.

Comment 9: Conclusion should include future recommendations

Response:  The future recommendations has been added from line number 812 to 820.

Reviewer 3 Report

Some comments to the manuscript submitted for review:

1. The authors in the "Methodology" section wrote that the information about genus Cirsium comes from "some of the unpublished data". What data did the authors have in mind?

2. Were phenolic acids identified only in C. canum and C. plaustre? They could not be identified in another representative of this species?

3. Table 4 should be divided into fragments, and each fragment should be placed respectively under the discussed biological activity, i.e. under sections 9.1, 9.2, etc.

4. Flavonoids or phenolic acids, which are often present in the chemical composition of plant extracts, are often responsible for their antioxidant properties. Could the authors indicate in section 9.2 the compounds responsible for the discussed biological activity?

5. The names of the journals in the "References" section should be written in accordance with the requirements for Authors.

Author Response

Comment 1: The authors in the "Methodology" section wrote that the information about genus Cirsium comes from "some of the unpublished data". What data did the authors have in mind?

Response:  The unpublished data means the information has been taken from online websites and not taken from research or review article. For instance

  1. Plants of the World Online: Cirsium Mill. Available online: http://www.plantsoftheworldonline.org/taxon/urn:lsid:ipni.org:names:30001899-2
  2. Creeping thistle facts and health benefits, Heal Benefits Times . (n.d.). https://www.healthbenefitstimes.com/creeping-thistle/ (accessed March 2, 2022).

Comment 2: Were phenolic acids identified only in C. canum and C. plaustre? They could not be identified in another representative of this species?

Response:  The phenolic acids are also identified in C. arvense and C. vulgare in addition to                 C. canum and C. plaustre . The text has been changed accordingly from line number 298 to 300 and references have also been added for the respective studies.

Comment 3: Table 4 should be divided into fragments, and each fragment should be placed respectively under the discussed biological activity, i.e. under sections 9.1, 9.2, etc.

Response:  The suggested changes has been done. Table 4 is fragmented into Table 4 to Table 6.

Comment 4: Flavonoids or phenolic acids, which are often present in the chemical composition of plant extracts, are often responsible for their antioxidant properties. Could the authors indicate in section 9.2 the compounds responsible for the discussed biological activity?

Response:  The compounds responsible for the biological activity in section 9.2 has been added from line number 430-438.

Comment 5: The names of the journals in the "References" section should be written in accordance with the requirements for Authors.

Response:  The name of the journals in Reference section are written according to the requirement of the Journal.

Round 2

Reviewer 1 Report

In terms of content, the work is good, but there are still significant flaws in the formatting and editing of the text.

And I still don't understand the difference between table 1 and 2. Why do the authors list table 2 when they have the same information in table 1, which is more detailed, and they always cite different sources.

The authors replied to all my comments that they made the changes, but it is not true.

And I still don't understand the difference between table 1 and 2. Why do the authors list table 2 when they have the same information in table 1, which is more detailed and they always cite different sources.

The authors replied to all my comments that they made the changes, but it is not true.

For some images to the captions at the bottom for others at the top - it should be uniform.

Why are some citation crossed out? (line 151, 265)

Edit bracket for quote 16 (line 103-104)

line 183 - the title should not be on one page and the text on the other.

Table headings are not uniform - somewhere with a space, somewhere without a space.

completely redo the formulas - non-uniform appearance, size, hard-to-read groups (drawn over each other), sugar formulas are different every time, again there are bonds in the formulas to which nothing is connected, non-uniform writing of groups (e.g.: OCH3 x O-CH3), labels for formulas also non-uniform - font, size, space from the formula.

Italics O in substance names – e.g.: Pectolinarigenin-7-O-Glucopyranoside, 6-Hydroxyluteolin 7-O-glucosid , writing capital and small letters in the names of substances - inconsistent - in my opinion, write everything in small letters

All this the author claims he did, but I don't see that being true.

Some citations in brackets have an extra space before the number (eg: lines 112, 136, 138166, tables ...)

citation line 204 - why is there a dash?

line 188 space before quote 47

Why do you write the names of substances in the text in small letters and in the formulas in capital letters? In the text, you write the names with a dash and not in the formulas, moreover, if it starts with, for example, alpha, then a lowercase letter is already written.

table 3 - on the other side, although the header is repeated, it is in the middle of the page and not at the top. Same page 15.

In some passages, there are unnecessary spaces between the text and others are missing.

picture 12 - why aren't the labels around the circle equally oriented?

table 8 - the table header cannot be on one page and the rest of the table on another.

Why do some citations have a DOI and not others? It should be uniform.

Although the text is interesting, the formal aspect significantly reduces its quality. For this reason, the level of the article does not reach the possibility of publication, because it is a shame not to check the basic rules of formatting and writing formulas. If the authors correct everything, I recommend publishing, otherwise I will have to reject the article.

Author Response

  1. Comment In terms of content, the work is good, but there are still significant flaws in the formatting and editing of the text. And I still don't understand the difference between table 1 and 2. Why do the authors list table 2 when they have the same information in table 1, which is more detailed, and they always cite different sources. The authors replied to all my comments that they made the changes, but it is not true. And I still don't understand the difference between table 1 and 2. Why do the authors list table 2 when they have the same information in table 1, which is more detailed and they always cite different sources.

Response: The table 2 have been deleted from the manuscript and key data is incorporated in the text.

  1. Comment: For some images to the captions at the bottom for others at the top - it should be uniform.

Response: All the captions has been provided at the bottom of the Figures.

  1. Comment: Why are some citation crossed out? (line 151, 265)

Response: We have added new references for the additional text which we have added in response to the comments of reviewer. It has changed the sequence of other references that’s why citations are crossed out in text. Now all the crossed citation are removed.

  1. Comment: Edit bracket for quote 16 (line 103-104)

Response: Needful done.

  1. Comment: line 183 - the title should not be on one page and the text on the other. Table headings are not uniform - somewhere with a space, somewhere without a space.

Response: The suggested changes has been done.

  1. Comment: completely redo the formulas - non-uniform appearance, size, hard-to-read groups (drawn over each other), sugar formulas are different every time, again there are bonds in the formulas to which nothing is connected, non-uniform writing of groups (e.g.: OCH3x O-CH3), labels for formulas also non-uniform - font, size, space from the formula.

Response: All the formulas are drawn with the help of chem draw software. Some times its difficult to separate closely spaced group like OH in case of sugars. Hope reviewer can understand the constrain. We tried our best to maintain the uniformity of the formulas but sometimes its difficult to arrange in the template provided by the publishers.   

  1. Comment: Italics Oin substance names – e.g.: Pectolinarigenin-7-O-Glucopyranoside, 6-Hydroxyluteolin 7-O-glucosid, writing capital and small letters in the names of substances - inconsistent - in my opinion, write everything in small letters. All this the author claims he did, but I don't see that being true.

Response: O- in the name of every compound where it appeared is converted to Italics, First word of all the phytochemical is capitalized and rest in formula is in running form with small letters.

  1. Comment: Some citations in brackets have an extra space before the number (eg: lines 112, 136, 138166, tables ...)

Response: Extra space before the number has been deleted.

  1. Comment: citation line 204 - why is there a dash?

Response: The dash has been left during formatting. It has been deleted.

  1. Comment: line 188 space before quote 47

Response: The space is adjusted before quoting the reference between 45-47.

  1. Comment: Why do you write the names of substances in the text in small letters and in the formulas in capital letters? In the text, you write the names with a dash and not in the formulas, moreover, if it starts with, for example, alpha, then a lowercase letter is already written.

Response: The names of all the phytoconstituents are written in with first letter capital both in formula and text. In case of any special character like alpha the first letter of the compound is written in capital.

  1. Comment: table 3 - on the other side, although the header is repeated, it is in the middle of the page and not at the top. Same page 15.

Response: The header of the table has been corrected.

  1. Comment: In some passages, there are unnecessary spaces between the text and others are missing.

Response: The space has been unified between the passages.

  1. Comment: picture 12 - why aren't the labels around the circle equally oriented?

Response: The picture 12 has been reformed.

  1. Comment: table 8 - the table header cannot be on one page and the rest of the table on another.

Response: The suggested changes has been done.

  1. Comment: Why do some citations have a DOI and not others? It should be uniform.

Response: The uniformity has been maintained among the references.
